# Simultaneous Control of *Staphylococcus aureus* and *Bacillus cereus* Using a Hybrid Endolysin LysB4EAD-LysSA11

**DOI:** 10.3390/antibiotics9120906

**Published:** 2020-12-14

**Authors:** Bokyung Son, Minsuk Kong, Yoyeon Cha, Jaewoo Bai, Sangryeol Ryu

**Affiliations:** 1Department of Food and Animal Biotechnology, Seoul National University, Seoul 08826, Korea; sonbk0722@gmail.com (B.S.); dianacha2006@naver.com (Y.C.); 2Department of Agricultural Biotechnology, Seoul National University, Seoul 08826, Korea; 3Department of Food Science and Technology, Seoul National University of Science and Technology, Seoul 01811, Korea; kongmin1@seoultech.ac.kr; 4Division of Applied Food System, Food Science & Technology, Seoul Women’s University, Seoul 01797, Korea; jwbai@swu.ac.kr; 5Research Institute of Agriculture and Life Sciences, Seoul National University, Seoul 08826, Korea; 6Center for Food and Bioconvergence, Seoul National University, Seoul 08826, Korea

**Keywords:** endolysin, protein engineering, hybrid protein, biocontrol agent

## Abstract

Bacteriophage endolysins have attracted attention as promising alternatives to antibiotics, and their modular structure facilitates endolysin engineering to develop novel endolysins with enhanced versatility. Here, we constructed hybrid proteins consisting of two different endolysins for simultaneous control of two critical foodborne pathogens, *Staphylococcus aureus* and *Bacillus cereus*. The full-length or enzymatically active domain (EAD) of LysB4, an endolysin from the *B. cereus*-infecting phage B4, was fused to LysSA11, an endolysin of the *S. aureus*-infecting phage SA11, via a helical linker in both orientations. The hybrid proteins maintained the lytic activity of their parental endolysins against both *S. aureus* and *B. cereus*, but they showed an extended antimicrobial spectrum. Among them, the EAD of LysB4 fused with LysSA11 (LysB4EAD-LyaSA11) showed significantly increased thermal stability compared to its parental endolysins. LysB4EAD-LysSA11 exhibited high lytic activity at pH 8.0–9.0 against *S. aureus* and at pH 5.0–10.0 against *B. cereus*, but the lytic activity of the protein decreased in the presence of NaCl. In boiled rice, treatment with 3.0 µM of LysB4EAD-LysSA11 reduced the number of *S. aureus* and *B. cereus* to undetectable levels within 2 h and also showed superior antimicrobial activity to LyB4EAD and LysSA11 in combination. These results suggest that LysB4EAD-LysSA11 could be a potent antimicrobial agent for simultaneous control of *S. aureus* and *B. cereus*.

## 1. Introduction 

Food poisoning outbreaks caused by bacterial pathogens are a major concern worldwide. In the United States, approximately 9.4 million foodborne illnesses with about 56,000 hospitalizations and 1300 deaths caused by major foodborne pathogens are reported every year [1]. In particular, 300,000 cases are caused by *Bacillus cereus* or *Staphylococcus aureus*, which are considered important bacteria in terms of infection frequency and seriousness of the disease [1]. A previous study demonstrated that *B. cereus* and *S. aureus* frequently contaminate foods such as milk or rice-based foods, and they can cause outbreaks in similar food products [2]. Moreover, the symptoms of food poisoning caused by *B. cereus* and *S. aureus* highly resemble other foodborne infections [3]. Due to these symptomatic similarities, *B. cereus* food poisonings are occasionally misdiagnosed as *S. aureus* intoxication [4]. Thus, simultaneous control of both *S. aureus* and *B. cereus* would be highly meaningful.

Since antibiotic-resistant bacteria have increased, phage endolysins have attracted attention as a promising alternative to antibacterial agents [5,6]. Most endolysins from the phages infecting Gram-positive bacteria have a modular structure, in which one or several enzymatically active domains (EAD) and a cell wall binding domain (CBD) are functionally separated by a short linker [7,8,9]. In the case of staphylococcal phage endolysins, they typically consist of a CHAP domain and an amidase domain as an EAD and a CBD [10]. The phage endolysins’ modular structure facilitates endolysin engineering to alter lytic activity, target specificity, protein solubility, and other properties [11]. A variety of protein engineering strategies have been attempted to optimize endolysins for specific applications [7,12]. Donovan et al. [13] reported that the full length and C-terminally truncated phage endolysin from *Streptococcus agalactiae* bacteriophage B30 was fused to the lysostaphin. These proteins degraded both streptococcal and staphylococcal cells, and their lytic activities were tested in milk. This group also created hybrid proteins with the endopeptidase of streptococcal phage λSA2 endolysin with the *Staphylococcus*-specific CBDs of either the endolysin LysK or lysostaphin. The altered CBDs conferred lytic activity against staphylococci and streptococci on the streptococcal enzyme [14].

Here, we created new hybrid endolysins targeting multiple pathogenic bacteria with two phage endolysins. Constructed hybrid endolysins acted independently and maintained lytic activity of their parental endolysins against both *S. aureus, B. cereus* and *Listeria monocytogenes,* indicating a successfully extended antimicrobial spectrum. In particular, LysB4EAD-LysSA11 showed higher thermal stability than their parental endolysins and potent antimicrobial activity against *S. aureus* and *B. cereus* in boiled rice. This hybrid endolysin would be a promising antimicrobial agent for the simultaneous control of *S. aureus* and *B. cereus*. Moreover, this approach will provide an opportunity to design multifunctional and highly specific antimicrobials, thereby helping reduce the incidence of multidrug-resistant bacteria.

## 2. Results and Discussion

### 2.1. Construction and Expression of the Hybrid Proteins

LysSA11 and LysB4 were selected for the construction of hybrid proteins with the goal of controlling *S. aureus* and *B. cereus* simultaneously. LysSA11 is an endolysin derived from the virulent *S. aureus* phage SA11 [15]. While most *S. aureus* endolysins are composed of three domains, LysSA11 is composed of two functional domains: a CHAP domain at its N-terminal region and a CBD at its C-terminal region, making it more suitable to be fused with other endolysins. More importantly, it has strong lytic activity against *S. aureus* strains. LysB4 is a *B. cereus* phage endolysin, which exhibited the enzymatic activity of an L-alanoly-D-glutamate endopeptidase on peptidoglycans [16]. LysB4 showed strong lytic activity against a broad range of pathogenic bacteria, including *Bacillus* species and *Listeria monocytogenes.* Using these two endolysins, we generated various types of hybrid endolysins. First, two full-length endolysins were connected by a helical linker, either LysSA11 at the N-terminal (LysSA11-LysB4) or LysB4 at the N-terminal (LysB4-LysSA11) (Figure 1A). Then, we created other hybrid endolysins with truncated endolysins with the potential to optimize their antimicrobial activity.

A previous study reported that the C-terminal domain of the endolysin PlyL encoded in the *B. anthracis* genome could inhibit the enzymatic activity of its EAD in the absence of the cognate target [17,18]. In this regard, we have constructed LysSA11EAD (1–159 amino acids) and LysB4EAD (1–176 amino acids), which are C-terminally truncated versions of LysSA11 and LysB4, respectively (Appendix A). The turbidity reduction assay revealed that the lytic activity of LysSA11EAD significantly decreased compared to that of full-length LysSA11 against *S. aureus* with the same molar concentration (0.3 µM). Even a higher amount of LysSA11EAD (1.0 µM) did not show comparable staphylolytic activity to 0.3 µM of LysSA11, suggesting that CBD is essential for the lytic activity of LysSA11. Similarly, LysPBC1 endolysin from *B. cereus*-infecting phage PBC1 showed stronger lytic activity against *B. cereus* than LysPBC1EAD [19]. On the other hand, LysB4EAD exhibited almost equal lytic activity to that of LysB4, indicating LysB4CBD may not be essential to the enzymatic activity of LysB4. We then constructed two more hybrid endolysins by connecting either LysSA11 at the N-terminal (LysSA11-LysB4EAD) or LysB4EAD at the N-terminal (LysB4EAD-LysSA11) (Figure 1A) via a helical linker. All constructed proteins were overexpressed in soluble form in *E. coli*, and purified proteins were visualized as a single band with the expected molecular mass on SDS-PAGE gel (Figure 1B).

### 2.2. Lytic Activity of the Hybrid Endolysins

The relative lytic activity of the hybrid endolysins was determined by measuring the decrease in optical density at 600 nm (OD_600_) of *S. aureus* (Figure 2A) and *B. cereus* (Figure 2B). The monitored OD_600_ values of *S. aureus* and *B. cereus* are presented in Appendix A and Appendix A, respectively. LysSA11-LysB4, LysB4-LysSA11, LysSA11-LysB4EAD, and LysB4EAD-LysSA11 were as active as their parental endolysins, indicating that constructed hybrid proteins acted independently and maintained the lytic activity of their parental endolysins. Interestingly, the staphylolytic activity of LysSA11 appeared to be affected by adding LysB4 or LysB4EAD to its N-terminus, resulting in a decrease in the lysis rate of LysB4-LysSA11and LysB4EAD-LysSA11. The results indicate that the orientation of the proteins to be fused can affect the lysis rate of hybrid proteins.

### 2.3. The Antibacterial Spectrum of the Hybrid Proteins

The antimicrobial activity of the hybrid proteins was evaluated against Gram-positive bacteria (Table 1). As previously reported, LysSA11 exhibited lytic activity against all staphylococcal strains tested and had no effect on *B. cereus* and other Gram-positive bacteria. Likewise, LysB4 was active against *B. cereus, B. subtilis,* and *L. monocytogens*, but not against staphylococcal strains. On the other hand, LysSA11-LysB4, LysB4-LysSA11, LysSA11-LysB4EAD, and LysB4EAD-LysSA11 appeared to have the same antibacterial spectrum, killing not only staphylococcal strains but also *B. cereus, B. subtilis*, and *L. monocytogens*. These results demonstrate that the antimicrobial spectrum of the constructed hybrid proteins was extended compared to their parental endolysins, and the hybrid proteins maintained the specificities of their parental endolysins.

### 2.4. Thermal Stability Determination

The thermal stability of the endolysins is an essential factor in the development of biocontrol agents. In general, most phage endolysins are not stable to heat and lose their activity above 50–60 °C [20]. The thermal stability assay showed that the lytic activity of LysSA11 significantly decreased after 30 min incubation at 45 °C against *S. aureus*. However, all hybrid proteins constructed in this study showed strong lytic activity at 45 °C and had a low activity even at 55 °C, demonstrating that the hybrid proteins had enhanced thermal stability compared to their parental endolysin, LysSA11 (Figure 3A). Meanwhile, the hybrid proteins showed a different thermal stability pattern against *B. cereus* from that against *S. aureus*. LysSA11-LysB4 and LysSA11-LysB4EAD retained only 20% of the lytic activity of LysB4 after heating at 55 °C. However, LysB4-LysSA11 and LysB4EAD-LysSA11 had similar thermal stabilities to LysB4, which was relatively stable at most of the temperatures tested. They also exhibited even higher lytic activity than LysB4 at 65 °C (Figure 3B). Consistent with our results, previous studies have found several hybrid proteins with improved thermal stability compared to their parental proteins, demonstrating that improvements are likely due to the α-helical linker [21,22,23]. The rigid structure of the helical linker provides enough space or specific physical adaptation between parental enzymes, enhancing the bifunctional activity of the hybrid protein. These results led us to assume that the thermal stability of hybrid endolysin might be affected by the inserted linker and the orientation of the parental endolysins to be fused. We selected LysB4EAD-LysSA11, which showed the highest lytic activity against both *S. aureus* and *B. cereus* at a high temperature, for further study.

### 2.5. Effect of pH and NaCl on the Lytic Activity of LysB4EAD-LysSA11

First, we evaluated the lytic activity of the hybrid proteins at different pH values. LysB4EAD-LysSA11 had more than 80% of its maximal activity at pH 8.0–9.0 against *S. aureus* (Figure 4A). In contrast, it was relatively stable under a broad range of pH between pH 5.4–9.0 against *B. cereus* (Figure 4B), showing more than 60% residual activity even at pH 5.4. These results could be attributed to the different behavior of *S. aureus and B. cereus* at various pH conditions [24]. *S. aureus* contains teichoic acids in the peptidoglycan (PG) layer, and the teichoic acids have ribitol phosphates, which make the *S. aureus* PG negatively charged. However, the positive charge from the free amino groups of D-Ala reduces the charge. According to a previous study, the *S. aureus* PG becomes neutral or even has a slightly positive charge at a low pH and is much more negatively charged at a high pH [24]. Therefore, in acidic conditions, LysB4EAD-LysSA11 (pI = 9.20) was more likely to be positively charged, and thus the lytic activity might decrease due to repelling the proteins and the *S. aureus* PG. Meanwhile, *B. cereus* contains uncharged polysaccharide branches rather than teichoic acid [25]. The presence of diaminopimelic acid and glutamic acid makes the *B. cereus* cell wall more negatively charged, allowing LysB4EAD-LysSA11 to be more active under a broad range of pH values compared to *S. aureus*.

Next, the influence of NaCl on the lytic activity of LysB4EAD-LysSA11 was determined. The lytic activity against *S. aureus* and *B. cereus* decreased more than 50% in the presence of 50 mM NaCl (Figure 4C,D), which must be improved for the development of LysB4EAD-LysSA11 as a biocontrol agent.

### 2.6. Antimicrobial Activity of LysB4EAD-LysSA11 in Boiled Rice

To confirm the potential of LysB4EAD-LysSA11 as a biocontrol agent, it was applied to boiled rice samples. The boiled rice was artificially contaminated with *S. aureus* and *B. cereus* to mimic the natural environment for bacterial contamination of foods. Rice-based foods are frequently implicated as a source of *S. aureus* and *B. cereus* food poisoning [26,27,28]. LysB4EAD-LysSA11 showed lytic activity in a limited salt concentration range, indicating that boiled rice is suitable for the activity test of LysB4EAD-LysSA11. The treatment of LysSA11 and LysB4EAD in combination (3.0 μM each) reduced viable *S. aureus* cells to undetectable levels within 1 h (Figure 5A) and *B. cereus* cells by 1 log CFU/mL within 3 h (Figure 5B). These results demonstrate that the antimicrobial activity of LysB4EAD against *B. cereus* is less effective in food than in reaction buffer. A previous study explained that complex matrixes such as food products might reduce the activity of endolysins by restricting access to the target bacteria [29]. In this regard, it is conceivable that LysB4 may require CBD to facilitate access to target bacteria in boiled rice. Meanwhile, 3.0 μM of LysB4EAD-LysSA11 eliminated all *S. aureus* and *B. cereus* cells from boiled rice within 2 h and 1 h, respectively, indicating that the hybrid protein has strong antimicrobial activity against *S. aureus* and *B. cereus* (Figure 5C,D). In boiled rice, LysB4EAD-LysSA11 showed enhanced antimicrobial activity compared to LysB4EAD itself against *B. cereus*, indicating that LysSA11 may serve to help LysB4EAD exert its enzymatic activity. It has been reported that LysSA11 does not have a cell wall binding ability against *B. cereus* since *S. aureus* and *B. cereus* have different PG structures [15]. However, LysSA11 displayed a positive net charge under experimental conditions (pI value of LysSA11, 9.2), suggesting that it can help the fusion partner (LysB4EAD) break down the cell wall through weak interactions with negatively charged target bacteria. We also observed that LysB4EAD-LysSA11 had increased antimicrobial activity compared to a mixture of LysSA11 and LysB4, which were used for generating a hybrid endolysin (Appendix A). These results indicate the evident advantages of the protein hybrid, such as an expansion of the antimicrobial spectrum and activity when it was applied to food.

## 3. Materials and Methods

### 3.1. Bacterial Strains and Growth Conditions

All *Staphylococcus* strains were grown in tryptic soy broth (TSB; Difco, Detroit, MI, USA) at 37 °C. All *Bacillus*, *Listeria*, *Streptococcus* strains were grown in brain heart infusion (BHI; Difco) broth at 37 °C. Gram-negative bacteria were grown in LB broth at 37 °C. All tested bacteria were grown under aerobic conditions. *Escherichia coli* DH5α and BL21 (DE3) star strains were used in the cloning and expression of proteins, respectively. Baird-Parker agar plates with egg yolk tellurite (BPA; Difco) were used for selective enumeration of *S. aureus* and a *Bacillus cereus* selective agar plate with egg yolk emulsion and polymyxin B supplement (Oxoid) was used for selective enumeration of *B. cereus*.

### 3.2. Construction of Recombinant Proteins

LysSA11, LysB4, and LysB4EAD were connected by a helical linker (EAAAK)_4_ according to a previous study [30]. Hybrid proteins were constructed by an overlapping extension PCR described previously [31]. Plasmids and primers used in this study are listed in Table 2. The *lysSA11* gene was amplified from pET29b-LysSA11 [15], while *lysB4* and *lysB4EAD* genes were amplified from pET15b-LysB4 [16]. The two overlapping PCR fragments containing a helical linker were used for the second PCR step to generate the *lysB4-lysSA11* or *lysB4EAD-lysSA11* in both orientations. The resulting PCR products were inserted between the BamHI/SalI restriction enzyme sites of pET28a and all constructed plasmids were confirmed by DNA sequencing at Macrogen Inc. (Seoul, Korea). The cloned plasmids were transformed into *E. coli* BL21 (DE3).

### 3.3. Protein Expression and Purification

The protein expression was induced with 0.5 mM IPTG (isopropyl-β-d-thiogalactopyranoside) at OD_600_ = 0.7, followed by additional incubation for 20 h at 18 °C. Bacterial cells were suspended in lysis buffer (50 mM Tris-HCl, 300 mM sodium chloride, and 30% glycerol at pH 8.0) and were disrupted by sonication at a duty cycle of 25% and an output control of 5 (BRANSON ULTRASONICS, Danury, CT, USA). After centrifugation (20,000× *g*, 30 min), the supernatant passed through a Ni-nitrilotriacetic acid (NTA) superflow column (Qiagen GmbH, Germany), and purification of the recombinant proteins was performed according to the manufacturer’s instructions. The purified protein was stored at –80 °C until use after the buffer was changed to the storage buffer (50 mM Tris-HCl, 300 mM NaCl, and 30% glycerol; pH 8.0) using PD Miditrap G-25 (GE healthcare, Amersham, Buck, UK).

### 3.4. Lytic Activity Assay

The lysis of the hybrid endolysins was assessed by a turbidity reduction assay (Son et al., 2012). Bacterial cells grown in the exponential phase were re-suspended with the reaction buffer (20 mM Tris-HCl, pH 8.0). Then, purified proteins were added to the cell suspension at a final concentration of 0.3 µM, and the OD_600_ reduction of cells was measured over time at room temperature using a SpectraMax i3 multimode microplate reader at 600 nm. The relative lytic activity was calculated after 60 min as follows: (ΔOD600 test (endolysin added)—ΔOD600 control (buffer only))/initial OD600.

The antimicrobial spectrum was tested by plate lysis assay as previously described [15]. Briefly, 10 μL of diluted endolysin (10 pmol) was spotted onto a freshly prepared bacterial lawn on agar plates. Spotted plates were air-dried in a laminar flow hood for 15 min and incubated overnight at 37 °C.

### 3.5. Effect of pH and Temperature on the Endolysin Activity

For the temperature stability assay of the hybrid proteins, lytic activity was measured at 25 °C for 60 min after the enzyme was incubated at various temperatures (4–65 °C) for 30 min. To study the effect of temperature on the enzymatic activity of the fusion proteins, 0.3 µM of each protein was added into a target cell suspension, and the mixture was incubated at different temperatures (4–65 °C) for 60 min for the reaction. To test the effect of pH on the lytic activity of the hybrid proteins, 0.3 µM of each protein were added to the *S. aureus* RN4220 and *B. cereus* ATCC 21768 cells suspended in the following buffers: 50 mM sodium acetate (pH 4.5 and 5.4), 50 mM Tris-HCl (pH 6.5–8.0), 50 mM glycine (pH 9.0), and 50 mM N-cyclohexyl-3-aminopropanesulfonic acid (pH 10.0).

### 3.6. Antimicrobial Activity in Food Samples

The hybrid proteins were evaluated for their lytic activity in boiled rice because rice is frequently a source of *S. aureus* and *B. cereus* food poisoning [26,27]. The samples were prepared as described previously [19]. Briefly, sterilized instant rice was purchased at a local market and was heated using a microwave. The resulting cooked rice (10 g) was homogenized in 40 mL distilled water. Then, the slurry samples were artificially contaminated with MRSA CCARM 3089 (10^5^ CFU/mL) and *B. cereus* ATCC 21768 (10^4^ CFU/mL), followed by pre-incubating at 25 °C for 1 h to allow them to adapt to boiled rice samples. Enzymes were added to each sample at concentrations of 0, 0.3 and 3 µM, and the mixtures further were incubated at 25 °C for 4 h. Viable bacterial cells (CFU/mL) were counted every 1 h by plating each sample on a BPA for *S. aureus* and *B. cereus* selective agar plates for *B. cereus*.

## 4. Conclusions

We proposed an endolysin engineering strategy to control *S. aureus* and *B. cereus* simultaneously by constructing hybrid proteins consisting of LysSA11 and LysB4. The hybrid proteins showed an extended antimicrobial spectrum, maintaining the lytic activity of their parental endolysins against both *S. aureus* and *B. cereus*. Among the hybrid proteins tested, LysB4EAD-LysSA11 showed the most improved thermal stability compared to its parental endolysins, representing a great advantage of the protein hybrid. We also observed that LysB4EAD-LysSA11 had a more effective antimicrobial activity against *S. aureus* and *B. cereus* in a food matrix compared to mixture of two parental endolysins. The overall characteristics of LysB4EAD-LysSA11 are summarized in Table 3. These results suggest that LysB4EAD-LysSA11 could be a potent antimicrobial agent for simultaneous control of multiple pathogenic bacteria and this study will be helpful in designing highly specific but multifunctional antimicrobials.

## Figures and Tables

**Figure 1 antibiotics-09-00906-f001:**
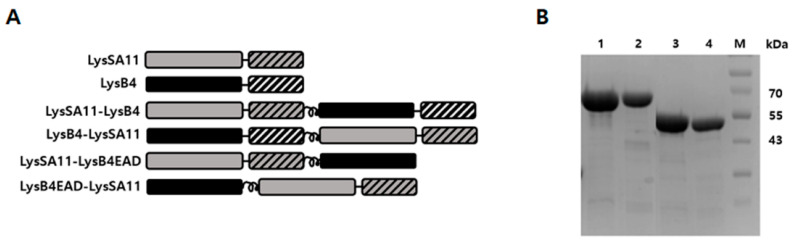
The modular structure of hybrid proteins and their lytic activity. (**A**) Schematic representation of LysSA11, LysB4, LysSA11-LysB4, LysB4-LysSA11, LysSA11-LysB4EAD, and LysB4EAD-LysSA11. LysSA11EAD: gray box; LysSA11CBD, gray diagonal stripe; LysB4EAD, black box; LysB4CBD, white diagonal stripe. (**B**) Purified LysSA11-LysB4, LysB4-LysSA11, LysSA11-LysB4EAD, and LysB4EAD-LysSA11 were loaded on an SDS-PAGE gel. Lane 1, purified LysSA11-LysB4 fraction; lane 2, purified LysB4-LysSA11 fraction; lane 3, purified LysSA11-LysB4EAD; lane 4, purified LysB4EAD-LysSA11 fraction; lane M, molecular weight marker.

**Figure 2 antibiotics-09-00906-f002:**
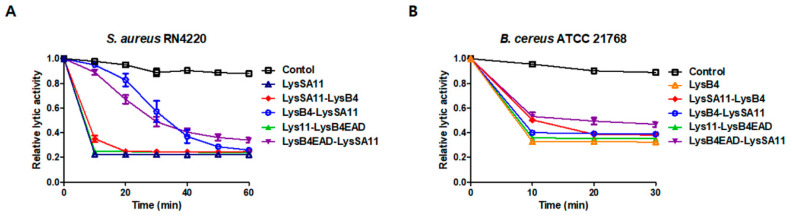
Lytic activity of the hybrid proteins. Relative lytic activity of LysSA11-LysB4, LysB4-LysSA11, LysSA11-LysB4EAD, and LysB4EAD-LysSA11 against (**A**) *S. aureus* RN4220 and (**B**) *B. cereus* ATCC 21768 was measured by monitoring a decrease in turbidity compared to their parental endolysins.

**Figure 3 antibiotics-09-00906-f003:**
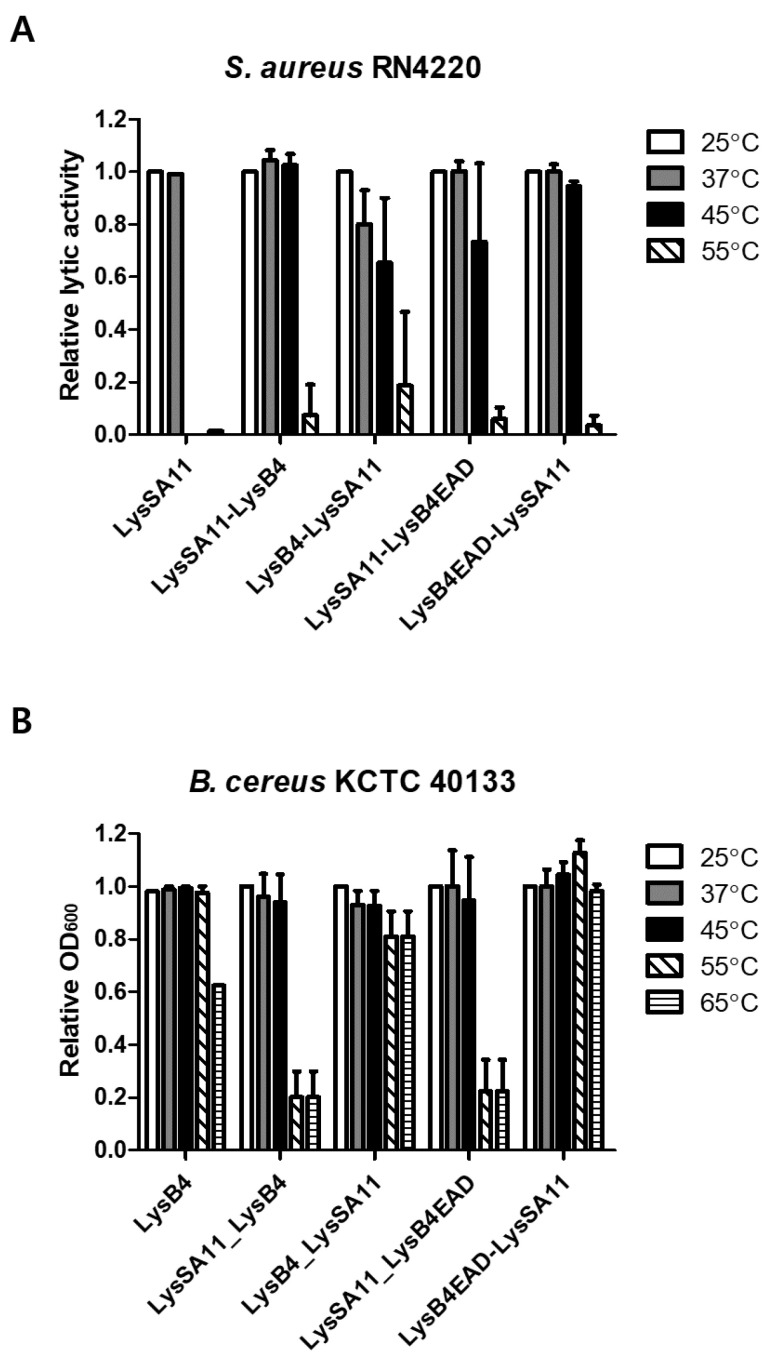
The thermal stability of the hybrid proteins. Equimolar concentrations (300 nM) of LysSA11, LysB4, LysSA11-LysB4, LysB4-LysSA11, LysSA11-LysB4EAD, and LysB4EAD-LysSA11 were incubated in reaction buffer at different temperatures for 30 min. Relative lytic activities against (**A**) *S. aureus* RN4220 and (**B**) *B. cereus* ATCC 21768 were calculated using the activity of the enzyme stored at 25 °C, which showed the maximal activity.

**Figure 4 antibiotics-09-00906-f004:**
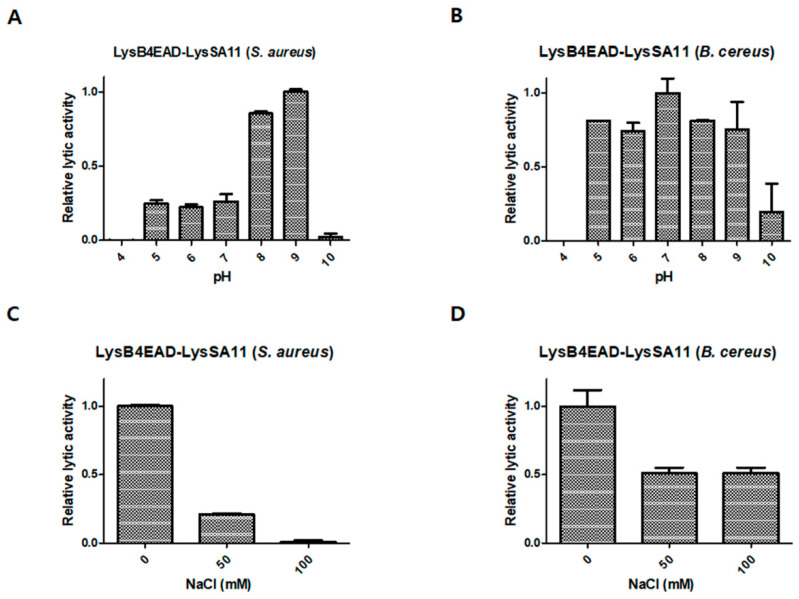
The effect of pH and NaCl on the lytic activity of LysB4EAD-LysSA11. The lytic activity of LysB4EAD-LysSA11 against (**A**) *S. aureus* RN4220 and (**B**) *B. cereus* ATCC 21768 was evaluated under different pH conditions. The lytic activity of LysB4EAD-LysSA11 against (**C**) *S. aureus* RN4220 and (**D**) *B. cereus* ATCC 21768 was evaluated at different NaCl concentrations.

**Figure 5 antibiotics-09-00906-f005:**
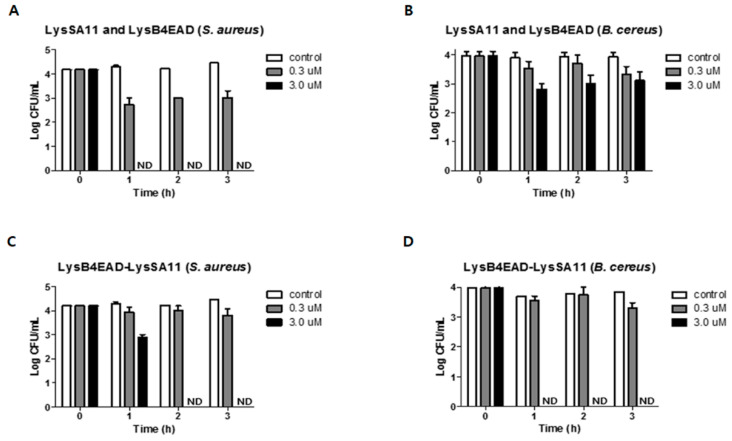
The antimicrobial activity of LysSA11 and LysB4EAD in combination and the hybrid endolysin LysB4EAD-LysSA11 in boiled rice contaminated simultaneously with *S. aureus* and *B. cereus*. The numbers of *S. aureus* RN4220 (**A**,**C**) and *B. cereus* ATCC 21768 (**B**,**D**) cells in boiled rice were counted after treatment with different concentrations (0.3 μM and 3.0 μM) of LysSA11 and LysB4EAD in combination (**A**,**B**), and LysB4EAD-LysSA11 (**C**,**D**). ND, not detected.

**Table 1 antibiotics-09-00906-t001:** The antibacterial spectra of the hybrid proteins.

		Lytic Activity ^a^					
Strains		LysSA11	LysSA11-LysB4	LysB4-LysSA11	LysSA11-LysB4EAD	LysB4EAD-LysSA11	LysB4
*S. aureus*	ATCC 33586	+	+	+	+	+	−
	ATCC 6538	+	+	+	+	+	−
	Newman	+	+	+	+	+	−
	RN4220	+	+	+	+	+	−
	ATCC 23235	+	+	+	+	+	−
	ATCC 29213	+	+	+	+	+	−
	ATCC 12600	+	+	+	+	+	−
	ATCC 33593	+	+	+	+	+	−
	ATCC 35983	+	+	+	+	+	−
	ATCC 13301	+	+	+	+	+	−
	CCARM 3793	+	+	+	+	+	−
	CCARM 3090	+	+	+	+	+	−
*S. hominis*	ATCC 37844	+	+	+	+	+	−
*S. saprophyticus*	ATCC 15305	+	+	+	+	+	−
*S. heamolyticus*	ATCC 29970	+	+	+	+	+	−
*S. capitis*	ATCC 35661	+	+	+	+	+	−
*S. warneri*	ATCC 10209	+	+	+	+	+	−
*S. xylosis*	ATCC29971	+	+	+	+	+	−
*S. epidermidis*	CCARM 3787	+	+	+	+	+	−
*B. cereus*	ATCC 21768	−	+	+	+	+	+
*B. cereus*	ATCC 27348	−	+	+	+	+	+
*B. subtilis*	168	−	+	+	+	+	+
*L. monocytogenes*	ATCC 19114	−	+	+	+	+	+

a +, Lysis; −, No lysis.

**Table 2 antibiotics-09-00906-t002:** Plasmids and primers used in this study.

Plasmids		
	Descriptions	References
pET28a	Kan^r^, T7 promoter, His-tagged expression vector	Novagen, Wisconsin, SA
pET15b-LysB4	pE15b with LSA12CBD	[16]
pET29b-LysSA11	pET29a with LSA97CBD	[15]
Primers (5′→3′)		
	Sequences	
B4EAD_HL_overl_R	TTT TGC CGC AGC TTC TTT TGC CGC AGC TTC TTT TGC CGC AGC TTC TTT TGC CGC AGC TTC TCC ACC TGT AGA GCC ACC TCC	
SA11EAD_Sal1_R	TTT GTC GAC TTG TAC CTC GTC TTT GAA ATT AGG	
SA11EAD_HL_overl_R	TTT TGC CGC AGC TTC TTT TGC CGC AGC TTC TTT TGC CGC AGC TTC TTT TGC CGC AGC TTC TTG TAC CTC GTC TTT GAA ATT AGG	
B4EAD_Sal1_R	TTT GTC GAC TCC ACC TGT AGA GCC ACC TCC	
BamH1_B4_F	AAA GGA TCC ATG GCA ATG GCA TTA CAA ACT T	
B4_HL_overl_R	TTT TGC CGC AGC TTC TTT TGC CGC AGC TTC TTT TGC CGC AGC TTC TTT TGC CGC AGC TTC TTT GAA CGT ACC CCA GTA GTT C	
SA11_Sal1_R	TTT GTC GAC TTT CCA GTT AAT ACG ACC CCA A	
BamH1_SA11_F	AAA GGA TCC ATG AAA GCA TCG ATG ACT AGA A	
SA11_HL_overl_R	TTT TGC CGC AGC TTC TTT TGC CGC AGC TTC TTT TGC CGC AGC TTC TTT TGC CGC AGC TTC TTT CCA GTT AAT ACG ACC CCA A	
HL_B4_overl_F	GAA GCT GCG GCA AAA GAA GCT GCG GCA AAA GAA GCT GCG GCA AAA GAA GCT GCG GCA AAA ATG GCA ATG GCA TTA CAA ACT TT	
B4_Sal1_R	TTT GTC GAC TTT GAA CGT ACC CCA GTA GTT C	

**Table 3 antibiotics-09-00906-t003:** Results summary.

Figure #	Target Bacteria	Observation
Figure 3A	*S. aureus*	LysB4EAD-LysSA11 is stable up to 45 °C while LysSA11 is stable up to 37 °C.
Figure 3B	*B. cereus*	LysB4EAD-LysSA11 is stable up to 65 °Cwhile LysB4 is stable up to 55 °C.
Figure 4A	*S. aureus*	LysB4EAD-LysSA11 is highly active at pH 8.0–9.0.
Figure 4B	*B. cereus*	LysB4EAD-LysSA11 is highly active at pH 5.0–9.0.
Figure 4C	*S. aureus*	The lytic activity of LysB4EAD-LysSA11 decreases in the absence of 50 mM NaCl.
Figure 4D	*B. cereus*	The lytic activity of LysB4EAD-LysSA11 decreases in the absence of 50 mM NaCl.
Figure 5A,C	*S. aureus*	3.0 μM of LysB4EAD-LysSA11 eliminates all bacterial cells in the boiled rice within 2 h while LysSA11 and LyB4EAD in combination (3.0 μM each) eliminates all bacterial cells within 1 h.
Figure 5B,D	*B. cereus*	3.0 μM of LysB4EAD-LysSA11 eliminates all bacterial cells in the boiled rice within 1 h while LysSA11 and LyB4EAD in combination (3.0 μM each) eliminates 1 log CFU/mL of bacterial cells within 3 h.

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
