# Peer review of "Simultaneous Control of Staphylococcus aureus and Bacillus cereus Using a Hybrid Endolysin LysB4EAD-LysSA11"

_antibiotics, 2020, doi:10.3390/antibiotics9120906_

Round 1

Reviewer 1 Report

The paper presents a very interesting approach to novel endolysin development, as antibacterial agents. The work is scientifically sound and simply and clearly presented.

As an optimization, for increasing the value of the article I would recommend  that the authors provide some form of graphical abstract, along with the written abstract. 

Also, concerning the results of the determinations of lytic activity at different temperatures and pH values, and antimicrobial activity in food samples it would be nice if the authors could produce a figure/table that briefly summarizes the comparative results at a glance. 

I therefore, consider this paper fit for publication in this journal.  

Reviewer 2 Report

The manuscript by B. Son et al reported construction of chimeric lysins from LysB4 and LysSA11 and their activities against S. aureus and B. cereus. The results are interesting and provide new routes for engineering lysins. However, the authors need to explain what could be the advantages and the disadvantages of fusing two lysins into one protein, compare with mixing two separate lysins? Other points are:

  1. Line 79: Please clarify S. aureus lysins consisit of three domains, not two domains? 
  2. Line 89-90: "...a C-terminally truncated version of endolysins LysSA11EAD.." should be "..a N-terminally truncated version of endolysins LysSA11EAD.."?
  3. Figure 5: It is better to give the results of decontamination using both lysins LysSA11 and LysB4 together. 
